# Optimized Separable Convolution: Yet Another Efficient Convolution Operator

## Abstract

The convolution operation is the most critical component in recent surge of deep learning research. Conventional 2D convolution needs $O(C^2K^2)$ parameters to represent, where $C$ is the channel size and $K$ is the kernel size. The amount of parameters has become really costly considering that these parameters increased tremendously recently to meet the needs of demanding applications. Among various implementations of the convolution, separable convolution has been proven to be more efficient in reducing the model size. For example, depth separable convolution reduces the complexity to $O(C \cdot (C + K^2))$ while spatial separable convolution reduces the complexity to $O(C^2K)$. However, these are considered ad hoc designs which cannot ensure that they can in general achieve optimal separation. In this research, we propose a novel and principled operator called *optimized separable convolution* by optimal design for the internal number of groups and kernel sizes for general separable convolutions can achieve the complexity of $O(C^{\frac{3}{2}}K)$. When the restriction in the number of separated convolutions can be lifted, an even lower complexity at $O(C \cdot \log(CK^2))$ can be achieved. Experimental results demonstrate that the proposed optimized separable convolution is able to achieve an improved performance in terms of accuracy-#Params trade-offs over both conventional, depth-wise, and depth/spatial separable convolutions.

## 1 Introduction

Tremendous progresses have been made in recent years towards more accurate image analysis tasks, such as image classification, with deep convolutional neural networks (DCNNs) (Krizhevsky et al., 2012; Srivastava et al., 2015; He et al., 2016; Real et al., 2019; Tan & Le, 2019; Dai et al., 2020). However, the complexity of state-of-the-art DCNN models has also become increasingly high. This can significantly deter their deployment to real-world applications, such as mobile platforms and robotics, where the resources and networks are highly constrained (Howard et al., 2017; Dai et al., 2020).

The most resource-consuming building block of a DCNN is the convolutional layer. There have been many previous works aiming at reducing the amount of parameters in the convolutional layer. Network pruning (Han et al., 2015) strategies are developed to reduce redundant parameters that are not sensitive to performances. Quantization and binarization (Gong et al., 2014; Courbariaux et al., 2016) techniques are introduced to compress the original network by reducing the number of bits required to represent each parameter. Low-rank factorization methods (Jaderberg et al., 2014; Ioannou et al., 2015) are designed to approximate the original weights using matrix decomposition. Knowledge distillation (Hinton et al., 2015) is applied to train a compact network with distilled knowledge from a large ensemble model. However, all these existing methods start from a pre-trained model. Besides, they mainly focus on network compression and have limited or no improvements in terms of network acceleration.

In this research, we study how to design a separable convolution to achieve an optimal implementation in terms of model size (representational complexity). Enabling convolution to be separable has been proven to be an efficient way to reduce the representational complexity (Sifre & Mallat, 2014; Howard et al., 2017; Szegedy et al., 2016). Comparing to the network compression related approaches, a well-designed separable convolution shall be more efficient in both storage and computation and shall not require a pre-trained model to begin with.

*Table 1.* A comparison of the number of parameters and computational complexity of the proposed optimized separable convolution and existing approaches. The proposed optimized separable convolution is much more efficient in both #Params and FLOPs. In this table, $C$ represents the channel size of convolution, $K$ is the kernel size, $H$ and $W$ are the output height and width, $g$ is the number of groups. "Vol. RF" represents whether the corresponding convolution satisfies the proposed volumetric receptive field condition.

| | Conventional Conv2D | Grouped Conv2D | Depth-wise Conv2D | Point-wise Conv2D | Depth Separable Conv2D | Spatial Separable Conv2D | Optimized Separable Conv2D ($N = 2$) | Optimized Separable Conv2D (Optimized $N$) |
|---|---|---|---|---|---|---|---|---|
| #Params | $C^2 K^2$ | $C^2 K^2 / g$ | $C K^2$ | $C^2$ | $C(C + K^2)$ | $2C^2 K$ | $2C^{\frac{3}{2}} K$ | $eC \log(CK^2)$ |
| FLOPs | $C^2 K^2 HW$ | $C^2 K^2 HW / g$ | $C K^2 HW$ | $C^2 HW$ | $CHW(C + K^2)$ | $2C^2 KHW$ | $2C^{\frac{3}{2}} KHW$ | $eCHW \log(CK^2)$ |
| Vol. RF | ✓ | ✗ | ✗ | ✗ | ✓ | ✓ | ✓ | ✓ |
| Note | – | – | $g = C$ | $K = 1$ | Depth-wise + Point-wise | $K^2 \rightarrow 2K$ | – | $e = 2.71828\ldots$ |

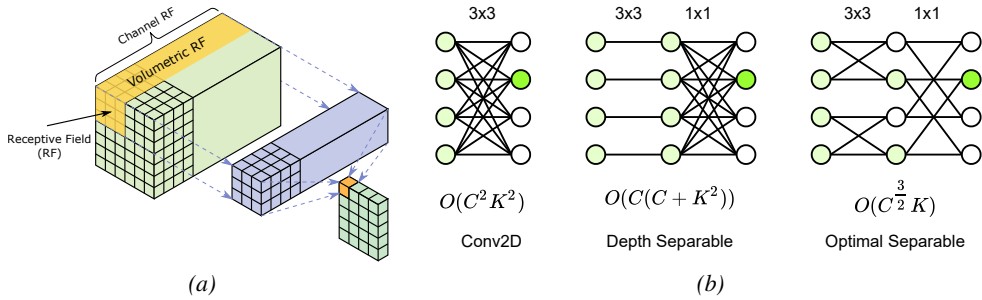

*(a)*          *(b)*

*Figure 1.* Volumetric receptive field and the proposed optimized separable convolution. (a) The volumetric receptive field (RF) of a convolution is the Cartesian product of its (spatial) RF and channel RF. (b) Illustrations of the channel connections for conventional, depth separable, and the proposed optimized separable convolutions. Optimized separable convolution is sparse-connected, whereas it can be efficiently implemented using a channel shuffle operation.

In the DCNN research, the two most well-known separable convolutions are depth separable (Sifre & Mallat, 2014) and spatial separable (Szegedy et al., 2016) convolutions. Both are able to reduce the complexity of a convolution. The representational complexity of a conventional 2D convolution is quadratic with two hyper-parameters: number of channels ($C$) and kernel size ($K$), and its representational complexity is actually $O(C^2 K^2)$. Depth separable convolution is constructed as a depth-wise convolution followed by a point-wise convolution, where depth-wise convolution is a group convolution with its number of groups $g = C$ and point-wise convolution is a $1 \times 1$ convolution. Spatial separable convolution replaces a $K \times K$ kernel with a $K \times 1$ and a $1 \times K$ kernel. Different types of convolutions and their complexities are summarized in Table 1. From this table, we can see that, for all convolutions, their computational complexities equal to the corresponding representational complexity times a constant. We can also verify that depth separable convolution has a complexity of $O(C \cdot (C + K^2))$ and spatial separable convolution has a complexity of $O(C^2 K)$.

Both depth and spatial separable convolutions follow an ad hoc design mode and are non-principled. They are able to reduce the complexity to some degree but normally cannot achieve an optimal separation. A separable convolution in general has three sets of hyperparameters: the internal number of groups, channel size, and kernel size of each separated convolution. Instead of setting these hyperparameters in an ad hoc (manual) fashion, we design a novel and principled (auto) scheme to achieve an optimal separation. The resulting separable convolution is called *optimized separable convolution* in this research. The proposed scheme in general performs better than the other convolution operator counterparts and it also enriches the separable convolution family.

To prevent the proposed optimized separable convolution from being degenerated, we assume that the internal channel size is in an order of $O(C)$ and propose the following *volumetric receptive field condition*. As illustrated in Fig. 1a, similar to the *receptive field (RF)* of a convolution which is defined as the region in the input space that a particular CNN's feature is looking at (or affected by) (Lindeberg, 2013), we define the *volumetric RF* of a convolution to be the volume in the input space that affects CNN's output. The volumetric RF condition requires a properly decomposed separable convolution to maintain the same volumetric RF as the original convolution before decomposition. Hence, the proposed optimized separable convolution will be equivalent to optimizing the internal number of groups and kernel sizes to achieve the target objective (measured in #Params) while

satisfying the proposed volumetric RF condition. Formally, the objective function is defined by Equation (2) under the constraints defined by Equations (3)-(6). The solution to this optimization problem will be elaborated in Section 2.

We shall show that the proposed optimized separable convolution can be represented with the order of $O(C^{\frac{3}{2}}K)$. This is at least a factor of $\sqrt{C}$ more efficient than the depth and spatial separable convolutions. The proposed optimized separable convolution is able to be generalized into an $N$-separable case, where the number of separated convolutions $N$ can be optimized further. In such a generalized case, an even lower complexity at $O(C \cdot \log(CK^2))$ may be achieved.

Extensive experiments have been carried out to demonstrate the effectiveness of the proposed optimized separable convolution over other alternatives, including conventional, depth-wise, depth and spatial separable convolutions (Fig. 3(c) and Fig. 4(c)). As further illustrated in Fig. 3 and Fig. 4, on the CIFAR10 and CIFAR100 datasets (Krizhevsky et al., 2009), the proposed optimized separable convolution achieves a better Pareto-frontier[1] than both conventional and depth separable convolutions using the ResNet (He et al., 2016) architecture. To demonstrate that the proposed optimized separable convolution generalizes well to other DCNN architectures, we adopt the DARTS (Liu et al., 2018) architecture by replacing the depth separable convolution with the proposed optimized separable convolution. The accuracy is improved from 97.24% to 97.67% with reduced representational complexity. On the ImageNet dataset (Deng et al., 2009), the proposed optimized separable convolution also achieves improved performance. For the DARTS architecture, the proposed approach achieves 74.2% top1 accuracy with only 4.5 million parameters. For MobileNet, the proposed approach achieves 71.1% top1 accuracy with only 3.0 million parameters.

## 2 THE PROPOSED APPROACH

### 2.1 CONVOLUTION AND ITS COMPLEXITY

A convolutional layer takes an input tensor $B_{l-1}$ of shape $(C_{l-1}, H_{l-1}, W_{l-1})$ and produces an output tensor $B_l$ of shape $(C_l, H_l, W_l)$, where $C_*$, $H_*$, $W_*$ are input and output channels, feature heights and widths. The convolutional layer is parameterized with a convolutional kernel of shape $(C_l, C_{l-1}, K_l^H, K_l^W)$, where $K_l^*$ are the kernel sizes, and the superscript indicates whether it is aligned with the features in height or width. In this research, we take $C_* = O(C)$, $H_* = O(H)$, $W_* = O(W)$, and $K_*^{H|W} = O(K)$ for complexity analysis. Formally, we have

$$B_l(c_l, h_l, w_l) = \sum_{c_{l-1}} \sum_{k_l^H} \sum_{k_l^W} B_{l-1}(c_{l-1}, h_{l-1}, w_{l-1}) \cdot F_l(c_l, c_{l-1}, k_l^H, k_l^W), \qquad (1)$$

where $h_l = h_{l-1} + k_l^H$ and $w_l = w_{l-1} + k_l^W$. Hence, the number of parameters for convolution is $C_l C_{l-1} K_l^H K_l^W$ and its representational complexity is $O(C^2 K^2)$. The number of FLOPs (multiply-adds) for convolution is $C_l H_l W_l \cdot C_{l-1} K_l^H K_l^W$ and its computational complexity is $O(C^2 K^2 HW)$.

For a group convolution, we have $g$ convolutions with kernels of shape $(C_l/g, C_{l-1}/g, K_l^H, K_l^W)$. Hence, it has $O(C^2 K^2 / g)$ parameters and $O(C^2 K^2 HW / g)$ FLOPs, where $g$ is the number of groups. A depth-wise convolution is equivalent to a group convolution with $g = C_* = C$. A point-wise convolution is a $1 \times 1$ convolution. A depth separable convolution is composed of a depth-wise convolution and a point-wise convolution. A spatial separable convolution replaces a $K \times K$ kernel with $K \times 1$ and $1 \times K$ kernels. Different types of convolutions are summarized in Table 1. From this table, their number of parameters and FLOPs can be easily verified. It can also be seen that, for a convolution, its representational complexity is equivalent to its computational complexity for up to a constant ($HW$).

### 2.2 RETHINKING CONVOLUTION AND THE VOLUMETRIC RECEPTIVE FIELD CONDITION

Separable convolution has been proven to be efficient in reducing the representational demand in convolution. However, existing approaches including both depth and spatial separable convolutions follow an ad hoc design and are non-principled. They are able to reduce the complexity to some extent but will not normally achieve an optimal separation. In this research, we shall design an

---

[1]In multi-objective optimization, a Pareto-frontier is the set of parameterizations (allocations) that are all Pareto-optimal. An allocation is Pareto-optimal if there is no alternative allocation where improvement can be made to one participant's well-being without sacrificing any other's. Here, Pareto-frontier represents the curve of the accuracies we are able to achieve for different #Params (or FLOPs).

efficient convolution operator capable of achieving the representational objective by optimal design of its internal hyper-parameters. The resulting operator is called *optimized separable convolution*.

The proposed optimized separable convolution is called principled as it optimizes the representational complexity under the following *volumetric receptive field condition*. As illustrated in Fig. 1a, the *receptive field* (RF) of a convolution is defined to be the region in the input space that a particular CNN's feature is affected by (Lindeberg, 2013). We define the *channel RF* to be the channels that affect CNN's output and define the *volumetric RF* to be the Cartesian product of the RF and channel RF of this convolution. The volumetric RF of a convolution actually represents the volume in the input space that affects CNN's output. The volumetric RF condition requires that *a properly decomposed separable convolution at least maintains the same volumetric RF as the original convolution before decomposition*. Hence, the proposed optimized separable convolution will be equivalent to optimizing its internal parameters while satisfying the volumetric RF condition. Formally, we shall have the objective function defined by Equation (2) and the volumetric RF constraints defined by Equations (3)-(6).

The volumetric RF of a convolution needs to be maintained for technical, conceptual, and experimental reasons. Technically, if we do not pose any restriction to a separable convolution, optimizing the representational complexity will resulting in a separable convolution being equivalent to a degenerated channel scaling operator[2]. The composition of such operators is not meaningful because the composition itself is equivalent to a single channel scaling operator. Conceptually, maintaining the volumetric RF encourages the fusion of channel information, which shall contribute to the good performance of a DCNN. In fact, all modern DCNNs are designed following this rule. Without this channel information exchange, the performance of a DCNN shall be significantly degraded (depth-wise vs depth separable convolutions in Section 3). Finally, the necessity of maintaining the volumetric RF is experimentally verified. We shall quantize the degree of necessity as overlap coefficient ($\gamma$) in Section 2.3 and elaborate the experimental results in Section 3.

## 2.3 Optimized Separable Convolution

In this section, for ease of simplicity, we first discuss the case of two-separable convolution. Suppose that the shape of the original convolutional kernel is $(C_{out}, C_{in}, K^H, K^W)$, where $C_{in}, C_{out}$ are the input and output channels, and $(K^H, K^W)$ is the kernel size. Let $C_1 = C_{in}$, and $C_3 = C_{out}$. For the proposed optimized separable convolution, we optimize the representational complexity as objective while maintaining the original convolution's volumetric RF. Formally, the representational demand of the proposed separable convolution is

$$f(g_1, g_2, C_2, K_*^{H|W}) = \frac{C_2 C_1 K_1^H K_1^W}{g_1} + \frac{C_3 C_2 K_2^H K_2^W}{g_2} \tag{2}$$

In order to satisfy the volumetric RF condition, the following three conditions need to be satisfied:

$$K_1^H + K_2^H - 1 = K^H \qquad (Receptive\ Field\ Condition) \tag{3}$$

$$K_1^W + K_2^W - 1 = K^W \tag{4}$$

$$g_1 \cdot g_2 \leq C_2/\gamma \Leftrightarrow \frac{C_1}{g_1} \cdot \frac{C_2}{g_2} \geq \gamma C_1 \qquad (Channel\ Condition) \tag{5}$$

$$\min(C_l, C_{l+1}) \geq g_l \qquad (Group\ Convolution\ Condition) \tag{6}$$

The channel condition (5) means the product $\frac{C_1}{g_1} \cdot \frac{C_2}{g_2}$ needs to occupy each node in the input channel $C_1 = C_{in}$ to maintain the volumetric receptive field. This is further explained for the channel condition general case (15) in Section 2.4. In order to study the necessity of the proposed volumetric RF condition, an overlap coefficient $\gamma$ is introduced to encourage channel information fusion. It can be verified that, if $\gamma \geq 1$, the channel RF shall be maintained, otherwise, it shall be not. By default, we set $\gamma = 1$ in this research unless the behavior of $\gamma$ is particularly concerned.

We have three sets of parameters: the number of groups $g_1$, $g_2$, the internal channel size $C_2$, and the internal kernel sizes $K_*^{H|W}$. In this research, we assume that the internal channel size $C_2$ is in an order of $O(C)$ and is preset according to a given policy. Otherwise, $g_1 = g_2 = C_2 = 1$ will be a trivial solution. This could lead the separable convolution to be over-simplified and not applicable

---

[2]From Table 1, let $g = C$ and $K = 1$, a convolution will have $C$ parameters and $CHW$ FLOPs. This is in fact a channel scaling operator.

in practice. Typical policies of presetting $C_2$ include $C_2 = \min(C_1, C_3)$ (normal architecture), $C_2 = (C_1 + C_3)/2$ (linear architecture), $C_2 = \max(C_1, C_3)/4$ (bottleneck architecture (He et al., 2016)), or $C_2 = 4\min(C_1, C_3)$ (inverted residual architecture (Sandler et al., 2018)).

The solution to the proposed optimized separable problem shall be given in Theorem 1 in Section 2.4. By setting $N = 2$ and $\gamma = 1$, we shall have

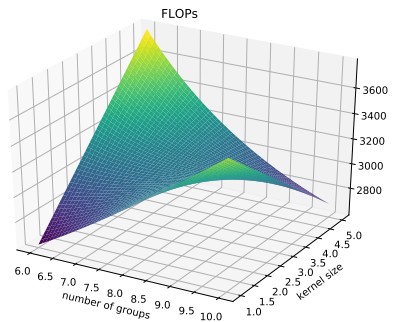

$$g_1 = \sqrt{\frac{C_1 C_2 K_1^H K_1^W}{C_3 K_2^H K_2^W}} \sim \sqrt{C}, \; g_2 = C_2/g_1 \quad (7)$$

$$(K_1^H, K_2^H) = (K^H, 1) \text{ or } (1, K^H) \quad (8)$$
$$(K_1^W, K_2^W) = (K^W, 1) \text{ or } (1, K^W) \quad (9)$$

and

$$\min f = 2 \cdot \sqrt{C_1 C_2 C_3 K_1^H K_1^W K_2^H K_2^W} = O(C^{\frac{3}{2}} K). \quad (10)$$

Figure 2. Given channels $C_1 = C_2 = C_3 = 64$, and kernel sizes $K^H = K^W = 5$ in Equation (2), by setting $f'(g_1) = 0$, $f'(K_1) = 0$. The solution $g_1 = 8$, $K_1 = 3$ is a saddle point.

One interesting fact is that if we set $g_2 = C_2/g_1$, $f'(g_1) = 0$, and $f'(K_1) = 0$, assume that kernel sizes aligned in height and width are equal, one can derive that $g_1$ is the same as Equation (7) and $K_1 = K_2 = \frac{K+1}{2}$.

Substituting them into Equation (10), one can get $f(g_1, K_1) = O(C^{\frac{3}{2}} K^2)$. This results in a higher complexity than $O(C^{\frac{3}{2}} K)$. In fact, the solution to $f'(g_1) = 0$ and $f'(K_1) = 0$ is a saddle point, which is illustrated in Fig. 2.

## 2.4 OPTIMIZED SEPARABLE CONVOLUTION (GENERAL CASE)

In this section, we shall generalize the proposed optimized separable convolution from $N = 2$ to an optimal $N$. For ease of analysis, we first introduce the notation *channels per group* $n_l = \frac{C_l}{g_l}$, which simply means: channels per group × number of groups = the number of channels.

**Theorem 1** (Optimized Separable Convolution: General Case). *Suppose that the shape of the original convolutional kernel is $(C_{out}, C_{in}, K^H, K^W)$. Let $C_1 = C_{in}$, and $C_{N+1} = C_{out}$. The representational demand of an $N$-separable convolution is*

$$f(\{g_*\}, \{K_*^{H|W}\}) = \sum_{l=1}^{N} \frac{C_{l+1} C_l K_l^H K_l^W}{g_l} \quad (11)$$

*or*

$$f(\{n_*\}, \{K_*^{H|W}\}) = \sum_{l=1}^{N} C_{l+1} n_l K_l^H K_l^W. \quad (12)$$

*Under the proposed volumetric RF condition, we will have:*

$$K_1^H + K_2^H + \cdots = K^H + (N-1) \qquad (Receptive\ Field\ Condition) \quad (13)$$
$$K_1^W + K_2^W + \cdots = K^W + (N-1) \quad (14)$$

$$n_1 \cdots n_N \geq \gamma C_1 \Leftrightarrow g_1 \cdots g_N \leq \frac{C_2 \cdots C_N}{\gamma} \qquad (Channel\ Condition) \quad (15)$$

$$n_l \geq \max(1, \frac{C_{l+1}}{C_l}) \Leftrightarrow g_l \leq \min(C_l, C_{l+1}). \quad (Group\ Convolution\ Condition) \quad (16)$$

*Assume that $C_* = O(C)$ and $K_*^{H|W} = O(K)$. The solution to this constrained optimization problem (the proposed optimized separable convolution problem) is given by*

$$n_l = \frac{\sqrt[N]{\gamma \Pi_{i=1}^{N+1} C_i \Pi_{i=1}^{N} K_i^H \Pi_{i=1}^{N} K_i^W}}{C_{l+1} K_l^H K_l^W} \sim \sqrt[N]{C} \quad (17)$$

$$K_{l_0}^H = K^H, \; K_l^H = 1 \; (l \neq l_0) \quad (18)$$
$$K_{l_1}^W = K^W, \; K_l^W = 1 \; (l \neq l_1) \quad (19)$$

*and its corresponding representational complexity is*

$$\min f(\{n_*\}, \{K_*^{H|W}\}) = O(N \gamma^{\frac{1}{N}} C^{1+\frac{1}{N}} K^{\frac{2}{N}}). \quad (20)$$

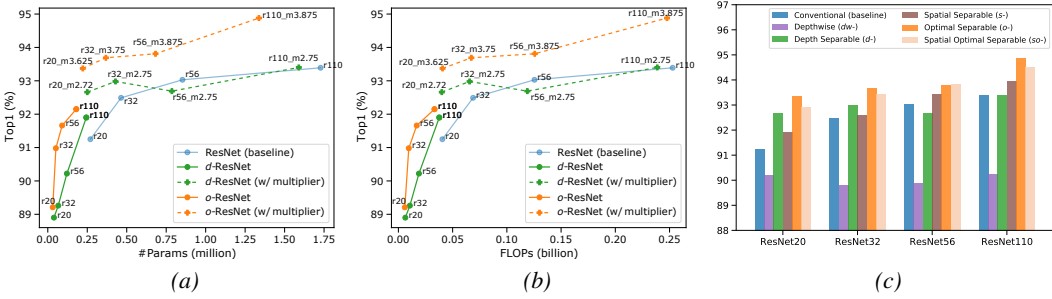

*(a)*        *(b)*        *(c)*

*Figure 3.* Experimental results on CIFAR10 for the ResNet architecture (best viewed in color). The proposed optimized separable convolution (*o*-ResNet) achieves improved (a) accuracy-#Params and (b) accuracy-FLOPs Pareto-frontiers than both the conventional (ResNet) and depth separable (*d*-ResNet) convolutions. (c) A comparison for performances of different convolution schemes.

*Table 2.* Experimental results on CIFAR10 for different overlap coefficients ($\gamma$). If $\gamma \geq 1$, the volumetric RF is maintained, otherwise it is not. Each row of *o*-ResNet is channel multiplied. When $\gamma < 1$, the performance hurts due to discourage of channel information fusion.

| Overlap Coef. ($\gamma$) | $\varepsilon$ (Depthwise) | 1/16 | 1/4 | 1 | 4 | 16 | 64 | $+\infty$ (Conventional) |
|---|---|---|---|---|---|---|---|---|
| Vol. RF | ✗ | ✗ | ✗ | ✓ | ✓ | ✓ | ✓ | ✓ |
| *o*-ResNet20 | 90.2 | 91.45 | 92.56 | **93.37** | 92.84 | 93.06 | 92.16 | 91.25 |
| *o*-ResNet32 | 89.82 | 92.31 | 92.8 | **93.69** | 93.65 | 93.03 | 93.07 | 92.49 |
| *o*-ResNet56 | 89.88 | 92.71 | 92.88 | **93.81** | 93.8 | 93.49 | 92.73 | 93.03 |
| *o*-ResNet110 | 90.26 | 94.04 | 94.85 | **94.88** | 94.83 | 94.41 | 93.95 | 93.39 |

*Furthermore, if the number of separated convolutions N can be optimized, we will have*

$$N = \log(\gamma C K^2) \tag{21}$$

*and*

$$\min f(\{n_*\}, \{K_*^{H|W}\}) = O(C \cdot \log(\gamma C K^2)). \tag{22}$$

In Theorem 1, we keep both notations $g_l$ and $n_l$. This is because, for the channel condition, it is intuitive to see $n_1 \cdots n_N \geq C_1$ means that the product of $n_1 \cdots n_N$ needs to occupy each node in the input channel $C_1 = C_{in}$. This is equivalent to the less intuitive condition $g_1 \cdots g_N \leq C_2 \cdots C_N$. Similarly, for the group convolution condition, $g_l \leq \min(C_l, C_{l+1})$ means the number of groups can not exceed the input and output channels of this group convolution, while $n_l \geq \max(1, \frac{C_{l+1}}{C_l})$ is less intuitive. A sketch of proof of Theorem 1 is given in Appendix A.

Equations (18) and (19) mean that one of the internal kernel sizes should take $K^H$ or $K^W$ and the remaining ones take 1. Hence, the proposed optimized separable convolution shall have a spatial separable configuration: a single kernel takes $(K^H, K^W)$ or two kernels take $(K^H, 1)$ and $(1, K^W)$. The implementation details of the proposed optimized separable convolution scheme is described in Algorithm 1 (Appendix B). Finally, the proposed optimized separable convolution is sparse connected. The hyperparameters of each separated convolution are given by Equations (17)-(19) and the proposed scheme can be efficiently implemented using a channel shuffle operation (Fig. 1b).

## 3 EXPERIMENTAL RESULTS

In this section, we carry out extensive experiments on benchmark datasets to demonstrate the effectiveness of the proposed optimized separable convolution scheme. In the proposed experiments, we use a prefix *dw*-, *d*-, *s*-, *o*- or *so*- to indicate that the conventional or depth separable convolutions in the baseline networks are replaced with depth-wise, depth separable (*dsep*), spatial separable, the proposed optimized separable (*osep*), or the proposed spatial optimized separable convolutions. In this research, we set the number of separated convolutions $N = 2$. The details of the training settings for the proposed experiments are described in Appendix C.

### 3.1 EXPERIMENTAL RESULTS ON CIFAR10

CIFAR10 (Krizhevsky et al., 2009) is a dataset consist of 50,000 training images and 10,000 testing images. These images are with a resolution of $32 \times 32$ and are categorized into 10 object classes. In

the proposed experiments, we use ResNet (He et al., 2016) as baselines and replace the conventional convolutions in ResNet with $dsep$ and $osep$ convolutions, resulting in $d$-ResNet and $o$-ResNet.

The proposed $osep$ scheme can significantly reduce the representational complexity. In Section 2, we state that this reduction factor can be $\sqrt{C}K$ in theory[3]. As illustrated by the *solid* lines in Fig. 3(a), the orange *solid* curve lies in a region with significantly smaller $x$-values than the blue *solid* curve. This indicates that $o$-ResNet shall have significantly less parameters than the ResNet baseline. For example, the 110-layered $o$-ResNet*110* has even fewer parameters (0.180 million vs

*Table 3.* Experimental results on CIFAR10 for DARTS. The proposed optimized separable convolution ($o$-DARTS) generalizes well to the DARTS architecture, and achieves improved accuracy with approximately the same FLOPs and fewer parameters. DARTS uses depth separable convolution and an optional $d$- is prefixed.

| Net Arch | #Params | FLOPs | Accuracy | Error Rate |
|---|---|---|---|---|
| | (million) | (billion) | (%) | (%) |
| ($d$-)DARTS (Liu et al., 2018) | 3.35 | 0.528 | 97.24% | 2.76% |
| $o$-DARTS | **3.25** | 0.572 | **97.67%** | **2.33%** |
| P-DARTS (Chen et al., 2019) | 3.43 | 0.532 | 97.50% | 2.50% |
| PC-DARTS (Xu et al., 2019) | 3.63 | 0.557 | 97.43% | 2.57% |
| GOLD-DARTS (Bi et al., 2020) | 3.67 | 0.546 | 97.47% | 2.53% |

0.270 million) than the 20-layered ResNet*20*, yet with noticeable higher accuracy (92.15% vs 91.25%). This demonstrates that the proposed $osep$ scheme could significantly reduce the representational complexity for convolutions. For $dsep$, this reduction factor is $\frac{1}{1/K^2+1/C}$, which is bounded by $K^2$. For $3 \times 3$ kernels, this reduction can be at most 9. Whereas for the proposed $osep$ scheme, no such bounds exist. The advantage of the proposed $osep$ scheme over $dsep$ is illustrated in Fig. 3(a) by the orange and green *solid* curves. From this, we can see the proposed $osep$ scheme is more efficient with smaller $x$-values. We further plot accuracy-FLOPs curves in Fig. 3(b) for reference, where similar conclusions can be drawn.

The proposed $o$-ResNets can have 10x-18x fewer parameters than the ResNet baselines in the proposed experiments. For fair comparisons, we introduce the channel multiplier in order to approximately match the #Params (or FLOPs)[4]. We use the suffix "_m<multiplier>" to indicate the channel multiplier. As illustrated in Fig. 3(a), from which we can see, the proposed $osep$ scheme is much more efficient than conventional convolutions. The orange curve, including both solid and dashed parts, achieved a better accuracy-#Params Pareto-frontier than the blue curve. Such representation efficiency could result in a more regularized network with fewer parameters to prevent over-fitting and possibly contribute to the final performance. In Fig. 3(a), we also present the $d$-ResNet curves in *dashed* green by replacing the conventional convolutions with $dsep$ convolutions. As can be seen, $d$-ResNet achieves good accuracy-#Params balances for small networks (e.g. $d$-ResNet20 and $d$-ResNet32), but performs comparable or no better than conventional convolutions for large ones (e.g. $d$-ResNet56 and $d$-ResNet110). In summary, the proposed $osep$ scheme achieves better accuracy-#Params (and also accuracy-FLOPs as illustrated in Fig. 3(b)) Pareto-frontiers than both conventional and $dsep$ convolutions.

**Other Conv2D Types:** Besides conventional and depth separable ($d$-) convolutions, we compare the proposed $osep$ ($o$-) scheme against the other convolution types, including depth-wise ($dw$-) and spatial separable ($s$-) convolutions. In the following, we shall omit the suffix of channel multiplier for simplicity, which shall be clear from the context. From Algorithm 1 (Appendix B), the proposed $osep$ scheme also has a spatial separable ($so$-) variant. A comparison of all these convolutions for the ResNet architecture is illustrated Fig. 3(c). From this figure, we can conclude that the proposed $osep$ scheme is more efficient than all other alternatives (the orange bar is highest).

**Channel Information Fusion:** We discuss more about $dw$-ResNet in Fig. 3(c). Recall from Table 1 that a depth separable convolution is a depth-wise convolution followed by a pointwise convolution. $dw$-ResNet allows no channel information exchange while $d$-ResNet does. Fig. 3(c) demonstrates that $dw$-ResNet performs much worse than $d$-ResNet. In fact, $dw$-ResNet is the only one that does not maintain the volumetric RF and performs worst of all these six convolution schemes. This

---

[3]For optimized separable, $\sqrt{C}K = \frac{C^2K^2}{C^{3/2}K}$. For depth separable, $\frac{1}{1/K^2+1/C} = \frac{C^2K^2}{C(C+K^2)} < K^2$.

[4]We match both #Params and FLOPs here. If this is not allowed, we approximately match for one and make sure the other not to exceed. In Appendix D, we present experimental results of matching #Params only in Fig. 5 and Fig. 6. The conclusions we reached in this Section is the same.

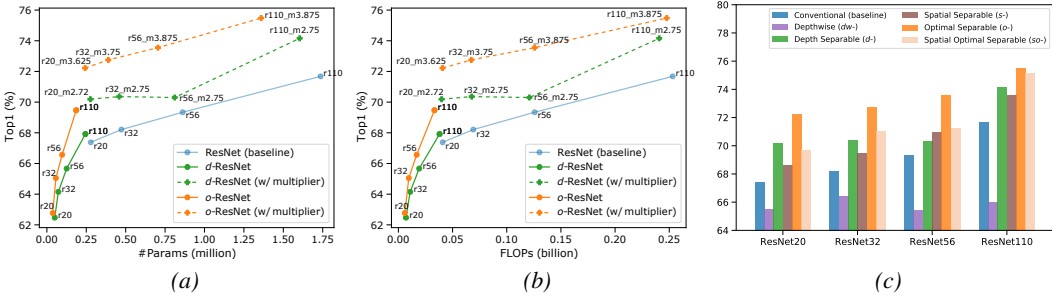

*Figure 4.* Experimental results on CIFAR100 for the ResNet architecture (best viewed in color). The proposed optimized separable convolution (*o*-ResNet) achieves improved (a) accuracy-#Params and (b) accuracy-FLOPs Pareto-frontiers than both the conventional (ResNet) and depth separable (*d*-ResNet) convolutions. (c) A comparison for performances of different convolution schemes.

*Table 4.* Experimental results on CIFAR100 for different overlap coefficients ($\gamma$). If $\gamma \geq 1$, the volumetric RF is maintained, otherwise it is not. Each row of *o*-ResNet is channel multiplied. When $\gamma < 1$, the performance hurts due to discourage of channel information fusion.

| Overlap Coef. ($\gamma$) | $\varepsilon$ (Depthwise) | 1/16 | 1/4 | 1 | 4 | 16 | 64 | $+\infty$ (Conventional) |
|---|---|---|---|---|---|---|---|---|
| Vol. RF | ✗ | ✗ | ✗ | ✓ | ✓ | ✓ | ✓ | ✓ |
| *o*-ResNet20 | 65.46 | 68.24 | 70.96 | 71.03 | **71.12** | 70.8 | 70.08 | 67.38 |
| *o*-ResNet32 | 66.42 | 70.59 | 71.05 | **72.75** | 70.89 | 71.91 | 70.76 | 68.21 |
| *o*-ResNet56 | 65.39 | 69.23 | 69.87 | **73.55** | 72.4 | 71.98 | 70.98 | 69.34 |
| *o*-ResNet110 | 65.98 | 73.30 | 74.00 | **75.48** | 74.74 | 73.62 | 73.01 | 71.68 |

suggests that channel information fusion could be critical for the good performance of a DCNN, and hence validates our proposed volumetric receptive field condition.

**Overlap Coefficient:** We carry out an ablation study on the overlap coefficient $\gamma$. For $\gamma \geq 1$, the volumetric RF is maintained, otherwise, it is not. From Table 2, we can see that, a good-performing $\gamma$ takes values $1 \leq \gamma \leq 4$ and $\gamma = 1$ achieves the best. It is reasonable to conjecture that, for $\gamma < 1$, the volumetric RF is not maintained and the channel information fusion is compromised, leading to bad performance. For $\gamma > 4$, the representation efficiency is also slightly lower. We argue that this is because the channel information has already been fused sufficiently. For larger $\gamma$, more overlap introduces more cost yet no additional fusion, hence the efficiency has been degraded accordingly. In Table 2, we also include the results for *dw*-ResNet and ResNet, as they can be roughly viewed as the limit cases of $\gamma$ to be infinitely small ($\varepsilon$) or infinitely large ($+\infty$). The ablation study on the overlap coefficient in Table 2 clearly demonstrates that we should satisfy the proposed volumetric RF condition.

**Generalization to DARTS:** To demonstrate that the proposed *osep* scheme generalizes well to other DCNN architectures, we adopt the DARTS (V2) (Liu et al., 2018) network as the baseline. The DARTS evaluation network has 20 cells and 36 initial channels, we increase the initial channels to 42 to match the FLOPs. By replacing the *dsep* convolutions in DARTS with the proposed *osep* convolutions, as illustrated in Table 3, the resulting *o*-DARTS improved the accuracy from 97.24% to 97.67%, but with fewer parameters (3.25 million vs 3.35 million). It is worth noting that it is very hard to significantly improve the DARTS search space. In Table 3, we also include three variants of DARTS, i.e. P-DARTS (Chen et al., 2019), PC-DARTS (Xu et al., 2019), and GOLD-DARTS (Bi et al., 2020), with more advanced search strategies for comparison. As can be seen, *o*-DARTS can achieve even higher accuracies than these advanced network architectures.

## 3.2 EXPERIMENTAL RESULTS ON CIFAR100

CIFAR100 (Krizhevsky et al., 2009) is a dataset consist of 50,000 training images and 10,000 testing images. These images are with a resolution of $32 \times 32$ and are categorized into 100 object classes. We carry out similar experiments on CIFAR100 as those on CIFAR10, from which similar conclusions can be drawn.

From Fig. 4(a) and (b), we can conclude that the proposed optimized separable convolution scheme achieves better accuracy-#Params and accuracy-FLOPs Pareto-frontiers than both conventional and depth separable convolutions. From Fig. 4(c), we can see that the proposed *osep* scheme is more

efficient than the other alternative Conv2D types, including depth-wise, spatial separable, and the proposed spatial optimized separable convolutions. In Fig. 4(c), $dw$-ResNet is the only one that does not maintain the volumetric RF and performs significantly worse than the other counterparts. In Table 4, the experimental results indicate the best overlap coefficient takes value $\gamma = 1$. These latter two observations also demonstrate the validity of the proposed volumetric RF condition.

### 3.3 EXPERIMENTAL RESULTS ON IMAGENET

We evaluate the proposed optimized separable convolution scheme on the benchmark ImageNet (Deng et al., 2009) dataset, which contains 1.28 million training images and 50,000 testing images.

#### 3.3.1 IMAGENET40

Because carrying out experiments directly on the ImageNet dataset can be resource- and time-consuming, we

*Table 5.* Experimental results on full ImageNet for the DARTS architecture. The proposed $o$-DARTS achieves 74.2% top1 accuracy with only 4.5 million parameters and the proposed $o$-MobileNet achieves 70.8% top1 accuracy with only 3.0 million parameters.

| Net Arch | #Params (million) | FLOPs (billion) | Top1 (%) | Top1 Error (%) |
|---|---|---|---|---|
| $(d\text{-})$DARTS (Liu et al., 2018) | 4.72 | 0.530 | 73.3% | 26.7% |
| $o$-DARTS | **4.50** | 0.554 | **74.2%** | **25.8%** |
| $(d\text{-})$MobileNet (Howard, 2017) | 4.20 | 0.575 | 70.6% | 29.4% |
| $o$-MobileNet | **3.00** | 0.564 | **71.1%** | **28.9%** |

resized all the images into $40 \times 40$ pixels. Due to space limitations, we present the experimental results on ImageNet40 in Appendix F and Table 7. We can conclude that the proposed $o$-ResNet achieved 4-5% (e.g. 49.97% vs 44.93% for 56-layer and 50.72% vs 46.74% for 110-layer) performance gains comparing against the ResNet baselines.

#### 3.3.2 FULL IMAGENET

Similar to the experiments on CIFAR10, we replace the $dsep$ convolutions in the DARTS (V2) network with the proposed $osep$ convolutions to demonstrate that the proposed approach is able to generalize to other network architectures. The experiment is carried out on the full ImageNet dataset. The DARTS evaluation network has 14 cells and 48 initial channels, we increase the initial channel size to 56 to match the original neural net. The resulting network is called $o$-DARTS. Experimental results are illustrated in Table 5. It can be seen that, with fewer parameters (4.50 million vs 4.72 million), the proposed $o$-DARTS network achieved higher accuracies in both top1 (74.2% vs 73.3%) and top5 (91.9% vs 91.3%) accuracies than the DARTS baseline. Finally, we replace the $dsep$ convolution in MobileNet (Howard et al., 2017) to the proposed $osep$ convolution. Using only 3.0 million parameters, the proposed $o$-MobileNet is able to achieve 71.1% top1 accuracy on the ImageNet dataset. This is a great gain comparing against the original MobileNet with 4.2 million parameters. We can conclude that the proposed $osep$ is able to achieve better accuracy-FLOPs and accuracy-#Params balances than $dsep$ convolutions.

## 4 CONCLUSIONS

In this paper, we have presented yet another novel convolution scheme called *optimized separable convolution* to improve efficiency. Conventional convolution took a costly complexity at $O(C^2 K^2)$. The proposed optimized separable convolution scheme is able to achieve its complexity at $O(C^{\frac{3}{2}} K)$, which is even lower than that of depth separable convolution at $O(C \cdot (C + K^2))$. Hence, the proposed optimized separable convolution *has the full potential to replace the usage of depth separable convolutions in a DCNN*. Examples considered include ResNet, DARTS, and MobileNet architectures. The proposed optimized separable convolution also has a spatial separable configuration. A generalized $N$-separable case can achieve better performance at $O(C \cdot \log(CK^2))$.

We believe the proposed optimized separable convolution also has a potential impact on the AutoML community. The proposed novel operator is able to increase the neural architecture search space. In a multi-objective optimization formulation, where both accuracy and #Params (or FLOPs) are optimized, we expect a more efficient network architecture can be discovered in the future using the proposed optimized separable convolution operator. In the future, we also expect to carry out experiments on more neural network architectures, e.g. EfficientNet, etc.

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

---

**Algorithm 1** The Algorithm for Optimized Separable Convolution

---

**Input:** Input channel $C_1 = C_{in}$, output channel $C_{N+1} = C_{out}$, kernel size $(K^H, K^W)$, number of separated convolutions $N$

**Optional Input**: internal kernel sizes (optional, preset), internal number of groups (optional, masked values), spatial separable (True or False), overlap coefficient ($\gamma = 1$).

**Output:** internal channel sizes $C_2, \cdots, C_N$, internal kernel sizes $K_1^{H|W}, \cdots, K_N^{H|W}$, internal number of groups $g_1, \cdots, g_N$

Calculate internal channel sizes $C_2, \cdots, C_N$ as $\min(C_{in}, C_{out})$, $\max(C_{in}, C_{out})/4$, or $4\min(C_{in}, C_{out})$, etc. according to a preset policy.

**if** internal kernel sizes $K_1^{H|W}, \cdots, K_N^{H|W}$ are not given **then**
    **if** spatial separable **then**
        Set $K_{\lfloor N/2 \rfloor}^H = K^H$, $K_{\lfloor N/2+1 \rfloor}^W = K^W$ and all other internal kernel sizes to 1.
    **else**
        Set $K_{\lfloor N/2 \rfloor}^{H|W} = K^{H|W}$ and all other internal kernel sizes to 1.
    **end if**
**end if**

Calculate internal channels per group $n_l$ according to $n_l = \frac{\sqrt[N]{\gamma \Pi_{i=1}^{N+1} C_i \Pi_{i=1}^N K_i^H \Pi_{i=1}^N K_i^W}}{C_{l+1} K_l^H K_l^W}$.

Let $g_l = \min(\lceil C_l/n_l \rceil, C_l, C_{l+1})$. If $C_l/n_l < 1$ or $C_l/n_l > \min(C_l, C_{l+1})$ for certain $l$, re-optimize $g_l$ with a masked number of groups by pre-setting $g_l = 1$ for $l \in \{l : C_l/n_l < 1\}$, $g_l = \min(C_l, C_{l+1})$ for $l \in \{l : C_l/n_l > \min(C_l, C_{l+1})\}$.
               ▷ Because $n_l \sim \sqrt[N]{C}$, for large channel sizes, we rarely need to re-optimize.

**Return** $C_2, \cdots, C_N$; $K_1^{H|W}, \cdots, K_N^{H|W}$; $g_1, \cdots, g_N$

---

## A    Sketch of Proof for Theorem 1

*Sketch of Proof for Theorem 1.* For Equation (12), after applying an arithmetic-geometric mean inequality, we can get

$$f(\{n_*\}, \{K_*^{H|W}\}) \geq N \sqrt[N]{\frac{C_1 C_2^2 \cdots C_N^2 C_{N+1} K_1^H \cdots K_N^H K_1^W \cdots K_N^W}{g_1 \cdots g_N}} \tag{23}$$

$$\geq N \sqrt[N]{\gamma C_1 \cdots C_{N+1} K_1^H \cdots K_N^H K_1^W \cdots K_N^W} \tag{24}$$

The equality holds if and only if $C_2 n_1 K_1^H K_1^W = \cdots = C_{N+1} n_N K_N^H K_N^W$. Let $n_l = \beta_l n_1$, where $\beta_l = \frac{C_2 K_1^H K_1^W}{C_{l+1} K_l^H K_l^W}$. Let $\beta = \Pi \beta_i$, we can solve $n_1 = \sqrt[N]{\frac{\gamma C_1}{\beta}} = \frac{\sqrt[N]{\gamma \Pi C_i \Pi K_i^H \Pi K_i^W}}{C_2 K_1^H K_1^W}$ and

$$n_l = \frac{\sqrt[N]{\gamma \Pi_{i=1}^{N+1} C_i \Pi_{i=1}^N K_i^H \Pi_{i=1}^N K_i^W}}{C_{l+1} K_l^H K_l^W} \sim \sqrt[N]{C}. \tag{25}$$

Note that the inequality (24) holds for arbitrary $K_*^{H|W}$. We need to further optimize $K_*^{H|W}$. Again, from the arithmetic-geometric mean inequality again, we can get $K_1^H \cdots K_N^H \leq (\frac{K_1^H + \cdots + K_N^H}{N})^N = (\frac{K^H + N - 1}{N})^N$ and the equality holds if and only if $K_1^H = \cdots = K_N^H = \frac{K^H + N - 1}{N}$. However, we want the inequality reversed, instead of finding the maximum of this product, we expect to find its minimum. This still gives us a hint, the maximum is achieved when the internal kernel sizes are as even as possible, so the minimum should be achieved when the internal kernel sizes are as diverse as possible. In the extreme case, one of the internal kernel sizes should take $K^H$ and all the rest takes 1, i.e. Equations (18) and (19). A formal proof of this claim can be derived. Hence, we have

$$f(\{n_*\}, \{K_*^{H|W}\}) \geq N \sqrt[N]{\gamma C_1 \cdots C_{N+1} K^H K^W} \tag{26}$$

$$= O(N \gamma^{\frac{1}{N}} C^{1+\frac{1}{N}} K^{\frac{2}{N}}). \tag{27}$$

By setting $f'(N) = 0$, we can derive that

$$N = \log(\gamma C K^2), \tag{28}$$

and

$$\min f(\{n_*\}, \{K_*^{H|W}\}) = eCHW \cdot \log(\gamma C K^2) \tag{29}$$

$$= O(CHW \cdot \log(\gamma C K^2)), \tag{30}$$

where $e = 2.71828...$ is the natural logarithm constant. $\qquad \square$

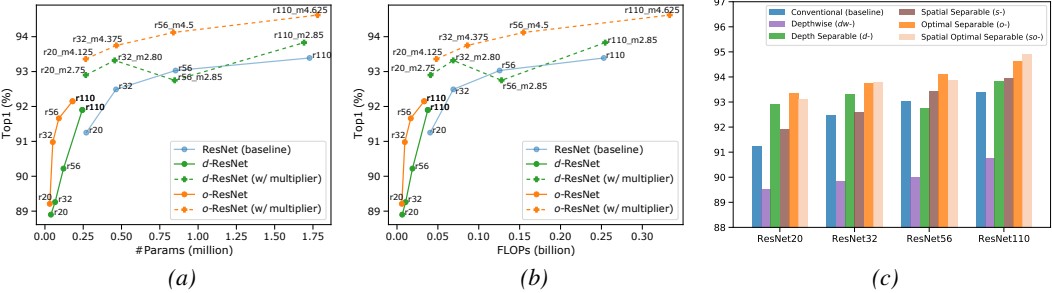

*Figure 5.* Experimental results on CIFAR10 for the ResNet architecture (best viewed in color, same as Fig. 3 except networks are channel multiplied to match #Params). The proposed optimized separable convolution (*o*-ResNet) achieves improved (a) accuracy-#Params and (b) accuracy-FLOPs Pareto-frontiers than both the conventional (ResNet) and depth separable (*d*-ResNet) convolutions. (c) A comparison for performances of different convolution schemes.

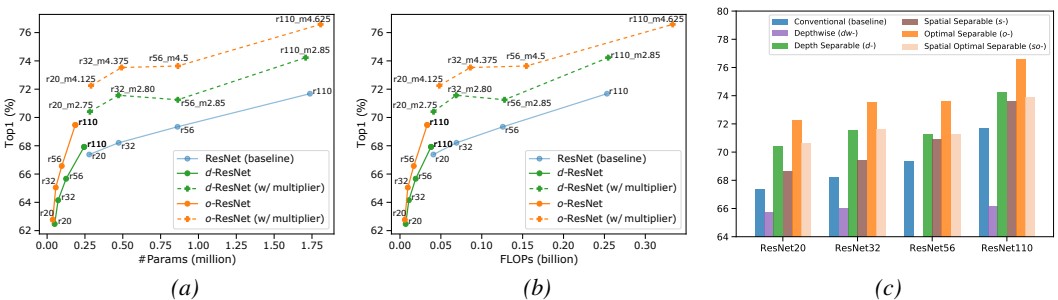

*Figure 6.* Experimental results on CIFAR100 for the ResNet architecture (best viewed in color, same as Fig. 4 except networks are channel multiplied to match #Params). The proposed optimized separable convolution (*o*-ResNet) achieves improved (a) accuracy-#Params and (b) accuracy-FLOPs Pareto-frontiers than both the conventional (ResNet) and depth separable (*d*-ResNet) convolutions. (c) A comparison for performances of different convolution schemes.

## B  THE PROPOSED OPTIMIZED SEPARABLE CONVOLUTION ALGORITHM

A detailed implementation of the proposed optimized separable convolution algorithm is illustration in Algorithm 1.

## C  TRAINING SETTINGS

**Experiments on CIFAR10 and CIFAR100 for the ResNet architecture**   The images are padded with 4 pixels and randomly cropped into $32 \times 32$ to feed into the network. A random horizontal flip with a probability of 0.5 is also applied. All the networks are trained with a standard SGD optimizer for 200 epochs. The initial learning rate is set to 0.1, with a decay of 0.1 at the 100 and 150 epochs. The batch size is 128. A weight decay of 0.0001 and a momentum of 0.9 are used.

**Experiments on CIFAR10 for the DARTS architecture**   We follow the same training settings in (Liu et al., 2018): the network is trained with a standard SGD optimizer for 600 epochs with a batch size of 96. The initial learning rate is set to 0.025 with a cosine learning rate scheduler. A weight decay of 0.0003 and a momentum of 0.9 are used. Additional enhancements include cutout, path dropout of probability 0.2, and auxiliary towers with weight 0.4.

**Experiments on ImageNet40 for the ResNet architecture**   Each network is trained with a standard SGD optimizer for 20 epochs with the initial learning rate set to 0.1, and a decay of 0.1 at the 10 and 15 epochs. The batch size is 256, the weight decay is 0.0001 and the momentum is 0.9.

**Experiments on full ImageNet for the DARTS architecture**   We follow the training settings in (Chen et al., 2019) for multi-GPU training: the images are random resized crop into $224 \times 224$

*Table 6.* Experimental results on CIFAR10 for the ResNet with inference time on a Windows 10 Intel i5-8250 CPU or Nvidia GeForce RTX 2080 Ti GPU.

| Net Arch | Channel Multiplier | #Params (million) | FLOPs (billion) | Accuracy (%) | CPU Infer Time (s) | GPU Infer Time (s) |
|---|---|---|---|---|---|---|
| ResNet20 | - | 0.270 | 0.04055 | 91.25 | 0.0310 | 0.0057 |
| $o$-ResNet20 | 3.625 | 0.221 | 0.04070 | 93.37 | 0.0468 | 0.0120 |
| $d$-ResNet20 | 2.72 | 0.250 | 0.0400 | 92.66 | 0.0468 | 0.0060 |
| ResNet32 | - | 0.464 | 0.06886 | 92.49 | 0.0469 | 0.0067 |
| $o$-ResNet32 | 3.75 | 0.367 | 0.06726 | 93.69 | 0.0937 | 0.0185 |
| $d$-ResNet32 | 2.75 | 0.429 | 0.06565 | 92.98 | 0.1154 | 0.0092 |
| ResNet56 | - | 0.853 | 0.12548 | 93.03 | 0.0938 | 0.0101 |
| $o$-ResNet56 | 3.875 | 0.682 | 0.12574 | 93.81 | 0.1562 | 0.0330 |
| $d$-ResNet56 | 2.75 | 0.786 | 0.11890 | 92.69 | 0.1875 | 0.0181 |
| ResNet110 | - | 1.728 | 0.25289 | 93.39 | 0.1563 | 0.0178 |
| $o$-ResNet110 | 3.875 | 1.337 | 0.24763 | 94.88 | 0.3216 | 0.0671 |
| $d$-ResNet110 | 2.75 | 1.590 | 0.23870 | 93.40 | 0.3462 | 0.0317 |

patches with a random scale in [0.08, 1.0] and a random aspect ratio in [0.75, 1.33]. Random horizontal flip and color jitter are also applied. The network is trained from scratch for 250 epochs with batch size 1024 on 8 GPUs. An SGD optimizer with an initial learning rate of 0.5, a momentum of 0.9, and a weight decay of 3e-5. The learning rate is decayed linearly after each epoch. Additional enhancements include label smoothing with weight 0.1 and auxiliary towers with weight 0.4.

**Experiments on full ImageNet for the MobileNet architecture**   The images are random resized crop into $224 \times 224$ patches with a random scale in [0.08, 1.0] and a random aspect ratio in [0.75, 1.33]. Random horizontal flip is also applied (no color jitter). The network is trained from scratch for 200 epochs with batch size 1024 on 8 GPUs. An SGD optimizer with an initial learning rate of 0.064, a momentum of 0.9, and a weight decay of 1e-5. The learning rate is decayed with a rate of 0.973 for every 0.8 epoch.

## D    EXPERIMENTAL RESULTS FOR MATCHING #PARAMS

In this section, we report experimental results of matching #Params only, instead of matching both #Params and FLOPs. In Fig. 5 and Fig. 6, we illustrate the experimental results on CIFAR10 and CIFAR100 for the ResNet architecture. As can be seen, the observations in these two figures are consistent with those in Fig. 3 and Fig. 4. Hence, the conclusions we reached in Section 3.1 and Section 3.2 are not affected. Please refer to these two sections for a detailed description of the experimental results.

## E    INFERENCE TIME FOR THE PROPOSED OPTIMIZED SEPARABLE CONVOLUTION

In this research, we focus on the representational efficiency of the proposed optimized separable scheme. The representational complexity is measured with the number of parameters (#Params) and is hardware-independent. For the proposed experiments, we included both #Params and FLOPs. In this section, we further report the wall-clock inference time of the proposed optimized separable convolution scheme for reference reasons. It is important to keep in mind that FLOPs measures the theoretical speed we are able to achieve. The wall-clock time reported in this section is hardware dependent. Slowness can occur due to an inefficient hardware implementation.

From Table 6, we can see that, under approximately the same FLOPs, both $o$-ResNet and $d$-ResNet take a longer inference time than conventional ResNet. This is because the current implementation of grouped convolution in PyTorch is not optimized. From Table 6, we can also conclude that, under approximately the same FLOPs, $o$-ResNet is slightly faster than $d$-ResNet (e.g. $o$-ResNet32 takes 0.0937s while $d$-ResNet32 takes 0.1154s) on a CPU and yet the former is about twice slower than the latter (e.g. $o$-ResNet32 takes 0.0185s while $d$-ResNet32 takes 0.0092s) on a GPU. The better wall-clock timing of the proposed $osep$ scheme over $dsep$ on a CPU may suggest that it also has

*Table 7.* Experimental results on ImageNet40 for the ResNet architecture. The proposed optimized separable convolution (*o*-ResNet) achieves 4-5% performance gain over the ResNet baseline.

| Net Arch | Channel Multiplier | #Params (million) | FLOPs (billion) | Accuracy (%) | Error Rate (%) |
|----------|--------------------|-------------------|-----------------|--------------|----------------|
| ResNet20 | - | 4.58 | 0.162 | 40.28 | 59.72 |
| *o*-ResNet20 | 5.375 | 5.13 | 0.160 | **44.94** | **55.06** |
| ResNet32 | - | 7.68 | 0.275 | 42.98 | 57.02 |
| *o*-ResNet32 | 5.75 | 7.78 | 0.278 | **47.88** | **52.12** |
| ResNet56 | - | 13.88 | 0.502 | 44.93 | 55.07 |
| *o*-ResNet56 | 6.0 | **12.55** | 0.497 | **49.97** | **50.03** |
| ResNet110 | - | 27.83 | 1.012 | 46.74 | 53.26 |
| *o*-ResNet110 | 6.25 | **23.79** | 1.027 | **50.72** | **49.28** |

advantages for ARM CPU architectures. Hence, the proposed *osep* scheme could be more efficient for mobile applications.

There are good reasons for the slowness of the proposed *osep* on a GPU. In fact, there are two extra copies of blobs in our current Python implementation of the proposed *osep* convolution (one for group convolution if the number of groups does not divide the input or output channels, and the other one for the channel shuffle operation). These two extra copies of blobs can actually be avoided for an efficient implementation. However, optimizing this code shall require a careful tweak of the CUDNN library. It is known that on a GPU, the memory access cost can dominate over the computational cost (Ma et al., 2018). Hence, the slowness of the proposed *osep* scheme shall occur. On a CPU, the computational cost dominates over the memory access cost. Hence, the proposed *osep* is faster than *dsep*. In the future, we expect the bottleneck of memory access for a GPU could be addressed and the wall-clock timing of the proposed *osep* scheme could be greatly sped up.

## F    EXPERIMENTAL RESULTS ON IMAGENET40

Because carrying out experiments directly on the ImageNet dataset can be resource- and time-consuming, we resized all the images into $40 \times 40$ pixels. A $32 \times 32$ patch is randomly cropped and a random horizontal flip with a probability of 0.5 is applied before feeding into the network. No extra data augmentation strategies are used. The baseline ResNet architecture is a modified version of that used on the CIFAR10 dataset, except that the channel sizes are set to be $4\times$ larger, the features are calculated on scales of [16, 8, 4], and the last fully-connected (FC) layer outputs 1000 categories for classification. We make this modification because the ImageNet dataset has significantly more training samples than the CIFAR10 dataset. Experimental results are illustrated in Table 7, as can be seen, by substituting conventional convolutions with the proposed optimized separable convolutions, the resulting *o*-ResNet achieved 4-5% (e.g. 49.97% vs 44.93% for 56-layer and 50.72% vs 46.74% for 110-layer) performance gains comparing against the ResNet baselines. This demonstrates that the proposed optimized separable convolution scheme is much more efficient. For *o*-ResNet56 and *o*-ResNet110, they also have fewer parameters which could contribute to a more regularized model. For *o*-ResNet20 and *o*-ResNet32, they have slightly more parameters because the last FC layer accounts for a great portion of overhead for 1000 classes.

## G    RELATED WORK

There have been many previous works aiming at reducing the amount of parameters in convolution. Network pruning (Han et al., 2015) strategies are developed to reduce redundant parameters that are not sensitive to performances. Quantization and binarization (Gong et al., 2014; Courbariaux et al., 2016) techniques are introduced to compress the original network by reducing the number of bits required to represent each parameter. Low-rank factorization methods (Jaderberg et al., 2014; Ioannou et al., 2015) are designed to approximate the original weights using matrix decomposition. Knowledge distillation (Hinton et al., 2015) is applied to train a compact network with distilled knowledge from a large ensemble model. However, all these existing methods start from a pre-trained model. Besides, they mainly focus on network compression and have limited or no improvements in terms of network acceleration.

Among various implementations of convolution, separable convolution has been proven to be more efficient in reducing the representational demand. Depth separable convolution is explored extensively in modern DCNNs (Howard et al., 2017; Sandler et al., 2018; Howard et al., 2019; Liu et al., 2018; Tan & Le, 2019). It reduces the representational cost of a conventional convolution from $O(C^2 K^2)$ to $O(C \cdot (C + K^2))$. However, the proposed optimized separable convolution is even more efficient than depth separable convolution. It can be represented at $O(C^{\frac{3}{2}} K)$ and has the full potential to replace the usage of depth separable convolutions. A second advantage of the proposed optimized separable convolution is that it can be applied to fully connected layers if we view them as $1 \times 1$ convolutional layers, whereas depth separable convolution cannot. Further, depth separable convolution requires the middle channel size to be equal to the input channel size, whereas for the proposed optimized separable convolution, the middle channel size can be freely set.

Spatial separable convolution was originally developed to speed up image processing operations. For example, a Sobel kernel is a $3 \times 3$ kernel and can be written as $(1, 2, 1)^T \cdot (-1, 0, 1)$. Spatial separable will require 6 instead of 9 parameters while doing the same operation. Spatial separable convolution is also adopted in the design of modern DCNNs. For example, in (Szegedy et al., 2016), the authors introduce spatial separation to the GoogLeNet (Szegedy et al., 2015) architecture. For the proposed optimized separable convolution, there is also a spatial separable configuration.

In the body of literature, separable convolution is also referred as *factorized convolution* or *convolution decomposition*. In this research, the proposed scheme is called optimized separable convolution following the naming conventions of depth and spatial separable convolutions.

