# OpenReview forum: "Optimized Separable Convolution: Yet Another Efficient Convolution Operator"
_ICLR.cc/2022/Conference — ICLR 2022 Submitted_

### Official Review · Reviewer_yy2C · 2021-10-30

**Correctness:** 2
**Technical Novelty And Significance:** 2
**Empirical Novelty And Significance:** 2
**Recommendation:** 5
**Confidence:** 4

**Main Review:**

Pros.
1.	This paper introduced the new design in potential by generalizing the depth-separable convolution. The proposed method extend common separable convolutions to more generalized & improved separable convolution.
2.	It demonstrated that generalized separable methods surpassed previous convolution methods in performance with a small number of parameters compared with existing separable methods.
3.    They have studied concrete analysis for flop's complexities. This supports that their approach is optimal among group separable convolutions.

Cons.
1.	The arguments about ad hoc design are self-contradictory. The introduction section mentioned that "Instead of setting these hyperparameters in an ad hoc fashion, we design a novel and principled scheme." However, in the experiment section, it seems that optimal separable methods also require hyperparameter (namely, overlap coefficient) tuning to obtain the best results.
2.	The authors claimed that by introducing the proposed convolution, the model size could be reduced. However, we can observe from Table 3 that model size is reduced by 0.1M (approximately 3 percent). This is a bit incremental even they achieved 1% accuracy up.
3.	In appendix section E, we can see that the GPU inference time of the proposed method is slower than other methods. To explain such a disadvantage, the authors say that PyTorch's implementation of group convolution is inefficient. However, it is believed that whenever group convolution is implemented in an efficient manner when considering the architecture of GPU, one-by-one convolution is the most memory-efficient design choice compared to other design choices, including group convolution.


**Summary Of The Paper:**

This paper proposed an optimized version of depth separable convolution. Optimal separable convolution reduces the model size by replacing both depth-wise and point-wise convolution with group convolution. Furthermore, they allow overlapped channel between group convolution and this was swept to show ablation comparisons. The authors showed that the combination of group convolution methods outperformed the previous ones when the volumetric receptive field (RF) conditions were met.

**Summary Of The Review:**

Overall, I think the authors have done a great job in explaining solid calculation for their optimum. However, their results are not convinced in terms of real elapsed time (slower GPU times) and marginal improvements in gain on number of parameters. The structure and representation of their work is good but empirical evidence is not sufficient.

---

### Official Review · Reviewer_a8wG · 2021-11-01

**Correctness:** 3
**Technical Novelty And Significance:** 2
**Empirical Novelty And Significance:** 2
**Recommendation:** 5
**Confidence:** 5

**Main Review:**

(1) Even if we do not discuss the NAS-net methods, the author should compare with other efficient CNN backbone designs, not limit the scope to separable convolutions and handcrafted optimization, for example, the IGCV series [1],  the Shift operations [2][3], learning the group strategies [4].

(2) The author should consider more criteria for the "efficient" architecture design, for example, the exact inference speeds (which are more important but listed in the appendix), the memory and time consumption during training [5]. This paper only shows the theoretical #FLOPs and # parameters, however, there is a huge gap between #FLOPs/parameters and the practical inference speed [6], which may prevent the practical implementation for the CNN community.

(3) The proposed optimization method seems quite heuristic, a description of motivation or theoretical proof would make the paper much more impressive.

[1] Sun, Ke, et al. "Igcv3: Interleaved low-rank group convolutions for efficient deep neural networks." arXiv preprint arXiv:1806.00178 (2018).

[2] Wu, Bichen, et al. "Shift: A zero flop, zero parameter alternative to spatial convolutions." Proceedings of the IEEE Conference on Computer Vision and Pattern Recognition. 2018.

[3] Chen, Weijie, et al. "All you need is a few shifts: Designing efficient convolutional neural networks for image classification." Proceedings of the IEEE/CVF Conference on Computer Vision and Pattern Recognition. 2019.

[4] Huang, Gao, et al. "Condensenet: An efficient densenet using learned group convolutions." Proceedings of the IEEE conference on computer vision and pattern recognition. 2018.

[5] Wu, Shuang, et al. "Convolution with even-sized kernels and symmetric padding." Advances in Neural Information Processing Systems 32 (2019): 1194-1205.

[6] Ma, Ningning, et al. "Shufflenet v2: Practical guidelines for efficient cnn architecture design." Proceedings of the European conference on computer vision (ECCV). 2018.

**Summary Of The Paper:**

This work proposes an optimized separable convolution by optimal designs for the internal number of groups
and kernel sizes for separable convolutions, thereby achieving better trade-offs between the model complexity and task performances.
The authors have done experiments and demonstrated the superiority of proposed methods on typical image classification tasks such as  CIFAR and ImageNet, as well as the generalization scalability on NAS-nets such as DARTs.


**Summary Of The Review:**

Overall, the idea and content of this paper are quite clear, the main concerns from the reviewer are the novelty of the proposed method and practical implementation for the CNN community.

---

### Official Review · Reviewer_bjqr · 2021-11-03

**Correctness:** 4
**Technical Novelty And Significance:** 2
**Empirical Novelty And Significance:** 3
**Recommendation:** 5
**Confidence:** 4

**Main Review:**

==== Weaknesses ====

This work seems more to be a structured analysis for motivating spatial separable and grouped convolutions and a careful way for determining the groups.

It seems a bit misleading to call this operator optimal, since the problem described is not completely the general case for convolutions. The channels are fixed in the intermediate convolutions and there are many other factors that could be introduced, such as spatial grouping factors or dilation factors in all dimensions. However, analytically and empirically this does seem like a good guide for setting the hyperparameters for spatial, grouped convolutions.

There are recent related works that should be included in the paper that deal with determining the groups in convolutions. "Differential Learning to Group Channels ..." (ICCV19) discusses an end-to-end way to determine these group factor. It would be interesting to compare how an empirical method like this compares with your analytical method. "Fully Learnable Group Convolution ... "(CVPR19) proposes a similar fully learnable group convolution. I believe the CondenseNets (CVPR18) may also have a learned group convolution. I understand that your method is more than just determining the number of groups, but especially in the '-o' case, not '-so', it seems the major contribution is the group number.

Computational complexity does not seem to be the most important metric here, since complexity usually deals with limit behavior. Many of these values, especially g and K, and usually small.

It would be helpful to fill out the curves more on the CIFAR comparisons, either though more layers or wider networks.

Subfigures a and b on Figure 3 and 4 seem mostly redundant since as mentioned in the paper they should be multiples of HW from each other. Also, the saddle point figure does not add much to the paper.

It seems most results are '-o', not '-so', which I interpret as not having the factored spatial component. I could be wrong on this, so please correct me. In that case, then then both the first and second convolution would have the full spatial components (e.g., 3x3) and this seems fundamentally different than a 'dsep' convolution since the spatial aggregation is much higher. This is like a higher spatial gamma factor (as opposed to the channel gamma introduced in the conditions).

The evaluation section especially on ImageNet could use more modern efficient architectures, like MobileNetV2 or EfficientNetV2 or others. Also, the experiments are interesting comparing depthwise to optimized, but it seems like all the of architectures are custom.

The takeaway from Table 4 and the ablation study about channel fusion seems fairly well-documented already in papers like ShuffleNet (CVPR18), Unitary Group Convolutions (CVPR19), and Interleaved Group Convolutions (CVPR17). This last work actually proposes a two-factored convolution that may serve as a reasonable comparison for this optimized convolution.


==== Strengths ====

This paper is very applicable since most efficient models include some grouped or factored or sparse convolutions.

Discussion of the trivial or degenerate cases help build better understanding of the problem, e.g. degenerate scaling operation.

Table 1 was a useful summary of existing methods.

The appendix included useful discussions on inference time.


==== Questions ====

What are the tradeoffs for these factored convolutions? The more factored convolutions (i.e. as N increases) seem to decrease the representational capacity. When would N > 2 be useful?

I'm a little confused when the results including the spatially factored convolutions in the results and when they don't. This confusion comes from the first paragraph of Section 3, when it discusses 'o-' and '-so'.

AutoML was mentioned in the conclusion. How do these operators compare to those returned by a traditional AutoML procedure where the groups and K factors were learnable?

Do any architectures or previous work already combine spatial and group convolutions?


**Summary Of The Paper:**

The convolution operator is the fundamental unit of most modern DNNs. This paper summarizes the existing (efficient) convolution operators and formulates an optimization problem to choose a few important parameters for the convolution. The authors show that their convolution under certain constraints requires fewer parameters and operations compared to the commonly used operators. They evaluate their proposed operator on CIFAR10, CIFAR100, and on ImageNet.

**Summary Of The Review:**

I believe this paper provides an interesting analysis suggesting how to configure the hyperparameters in efficient convolutional operators. The novelty from my reading of the paper was mostly confined to the analysis and conclusions around group number. I thought there were some key areas of related works missing, and the evaluation section was promising but a bit underwhelming relying mostly on CIFAR and custom ResNet models. Overall, I would argue that this paper should be a borderline reject.

---

### Official Review · Reviewer_VokC · 2021-11-04

**Correctness:** 3
**Technical Novelty And Significance:** 3
**Empirical Novelty And Significance:** 3
**Recommendation:** 5
**Confidence:** 3

**Main Review:**

Pros:
- The paper proposes a new extension of the seperable convolution family that attempts to automatically select internal parameters that optimize model complexity and expressiveness.
- The paper provides interesting analysis.
- There's a definite use case in the field of AutoML for a block of this type. It would be interesting to see this evaluated.
- The work shows improvement in common Cifar and ImageNet benchmarks.
- Satisfying the volumetric receptive field criteria was an interesting addition.

Cons:
- The paper is overall somewhat difficult to follow and could benefit from improved explanations throughout.
- The paper should have discussed attributes of efficient models beyond just FLOPS and params, such as memory overhead, FLOP vs memory balancing (which seperable convolutions often suffer from), etc.
- The paper could benefit from more advanced network evaluations.

While I found the proposals by the authors to be appealing, I personally found the paper difficult to read, and would encourage the authors to further improve explanations of concepts and overall paper flow, as it was rather difficult to follow at times.

**Summary Of The Paper:**

The authors propose a new convolutional operation designed to optimize the internal number of groups, and kernel sizes for a given model size and performance tradeoff. The authors compare their results with similar seperable convolution methods, and evaluate their methods and show performance gains on common benchmarks.

**Summary Of The Review:**

Overall I find the work to propose an appealing convolutional operation, in that it attempts to satisfy optimal design for a given model size and complexity. I believe the paper was difficult to read and would benefit from improved explanations of concept and flow.

---

### Decision · Program_Chairs · 2022-01-20

**Decision:**

Reject

**Comment:**

The paper introduces a convolutional-like operator called optimized separable convolution, which scales well in the number of channels, C. The paper studies the classification performance of ResNet with the optimized separable convolution, and with other choices of the convolutional operators. The paper finds the introduced convolutional  operator to be more parameter efficient than competing operators, however only by a relatively small margin, and this also comes at the cost of computational performance.

The reviewers appreciated that the proposed convolutional operation is more parameter efficient and that any improvement in convolutional networks potentially benefits a large array of methods and tools. The reviewers  criticize that the operation only offers marginal gains at the cost of slower runtimes, and agree that the contribution is only marginally significant and therefore below the acceptance threshold.

I agree with the reasoning of the reviewers that the extra computational cost is not worth the marginal improvement, and therefore recommend to reject the paper. Also, the authors didn't respond to the comments of the reviewers and their reasonable questions.